

# A low-cost autonomous rover for polar science

Andrew O. Hoffman [1], Hans Christian Steen-Larsen[2], Knut Christianson[1], Christine Hvidberg[3]

[1]Department of Earth and Space Sciences, University of Washington, Seattle, Washington 98195, USA
5 [2]Geophysical Institute of Bergen and Bierknes Center for Climate Research, Bergen, Norway
[3]University of Copenhagen and Centre for Ice and Climate, Copenhagen, Denmark
*Correspondence to*: Andrew O. Hoffman (hoffmaao@uw.edu)

## Abstract

10 We present the developmental considerations, design, and deployment of an autonomous modular terrestrial rover for ice-sheet exploration that is inexpensive, easy to construct, and allows for instrumentation customization. Total construction cost for this rover is less than $3000, approximately one tenth the cost of existing platforms, and it can be built using facilities frequently available at academic institutions (machine shop, 3D printer, open-source hardware and software). Instrumentation deployed on this rover can be customized; the rover presented in this study was equipped with a dual-frequency GPS receiver and a 15 digital SLR camera for constructing digital elevation models using structure-from motion (SfM) photogrammetry. We deployed this prototype rover on the Northeast Greenland Ice Stream to map local variations in snow accumulation and surface topography. The rover conducted four autonomous missions based out of the East Greenland Ice Core Project (EGRIP) camp during July 2017, measuring surface elevation transects across the hazardous ice-stream shear margins. During these missions, the rover proved capable of driving over 20km on a single charge with a drawbar pull of 250N, sufficient to tow commercial 20 ground-penetrating radars. The rover also acquired photographs that were subsequently used to construct digital elevation models of a site monitored for spatiotemporal variability in snow accumulation, demonstrating adequate stability for high-resolution imaging applications. Due to its low cost, low-power requirements, and simple modular design, mass deployments of this rover design are practicable. Furthermore, operation of the rover in hazardous areas circumvents substantial expense and risk to personnel associated with conventional, crewed deployments. Thus, this rover is an investigatory platform that 25 enables direct exploration of polar environments considered too hazardous for conventional field expeditions.

## 1 Introduction

Despite the recent proliferation of autonomous platforms (airborne drones, extra-terrestrial rovers, and even autonomous vehicles), deployment of conventional autonomous vehicles in polar regions remains challenging due to the hostile environment, stringent low-power requirements, and limits on construction costs (compared to, for example, extra-terrestrial 30 rovers). Temperatures in polar regions routinely drop below -40°C, well below the temperature ratings of conventional electronics, and wind speeds can easily exceed 20 metres per second, inducing variable stresses on any large profile robotic platform. As sunlight is often completely absent for several consecutive months, rover designs that leverage solar power are only viable during the summer, precisely when there are often already field personnel capable of conducting similar



measurements. Although precipitation is low over the interior of the ice sheets, snow can drift into the chassis, where subsequent snowmelt due to heat from powered components can then damage electronic components designed to remain dry. Drifted snow also forms sastrugi and other large snow bedforms, introducing a complex and time-variable topography of variable gravitational resistance, which presents a challenge for any robotic assembly designed to drive across the surface of

the ice sheet. Because of these environmental challenges, very few autonomous robotic exploration platforms have been built and successfully deployed to survey the polar ice sheets. Of these systems, NASA's rover, *GROVER* (Trisca et al., 2013; Robertson et al., 2013), and Cold Regions Research and Engineering Lab platforms, *Yeti* and *Cool Robot,* are the only non-combustion rovers actively conducting polar science (Lever et al. 2013; Lever et al., 2006). The high fabrication cost of these rovers is, however, an impediment to the construction of multiple rovers that would lead to extensive deployments, as each

requires over $10,000 in materials alone. Less expensive, expendable rovers have been developed for military, police, and rescue operations (Weisbin et al., 1999; Maddux et al., 2006; U.S. Department of Defence, 2013), but these tactical autonomous robots have been deployed in highly variable urban terrain with instruments and drive systems designed for that specific application (Blitch,1998), and they are not well-suited for polar environments.

Here we present the design, construction, and deployment of a largely autonomous, low-cost, modular rover that can expand

ice-sheet science programs in duration and extent without straining budgets or exposing personnel to risk. The rover we built was intended to enhance the surface science capabilities at the East Greenland Ice Core Project (EGRIP) camp; thus, although designed for general operation in an ice-sheet interior, instrumentation was customized to conduct surface and near subsurface measurements, including SfM photogrammetry, aerosol and isotope sampling, and snow accumulation radar. Our simple, low-cost design makes the rover built at the Centre for Ice and Climate, hereafter termed the CIC rover, truly expendable and opens

up new possibilities for coordinated autonomous multi-rover surface science. In this paper, we detail the mechanical, software, and instrumentation considerations for the CIC rover, present the rover's performance during an exploratory field deployment on the northeast Greenland ice stream, and, finally, discuss the outlook for widespread construction and deployments of variants of the CIC rover for other applications.

## 1.1 Polar rover research

Early polar rover research for glaciological science has been well characterized by Lever and Delaney et al. (2012) and Ray et al. (2007). Together, these groups built the first two autonomous battery powered polar science rovers, both designed with the goal to image internal ice structure in Greenland and the Antarctic by pulling ground penetrating radar. The *Yeti* rover has been used successfully on traverses and can image and detect bridged crevasses in real time, warning traverse teams of immediate subsurface dangers along the route. The Yeti platform has collected more published data than any other rover platform even

with a limited 20 km mission range, but the platform has been underutilized in science investigations outside of englacial imaging along traverse routes because of its expense and finite range. The solar powered *Cool Robot* boasts significant software enabled autonomy, capable of traversing >1000 km across the ice sheets in the polar summer while communicating via iridium satellite with off-ice operators. Since the development of these two autonomous rovers, *Yeti* and *Cool Robot*, NASA built the


Goddard Remotely Operated Vehicle for Exploration and Research (GROVER), which has been used to image the near surface firn near Summit, Greenland at high vertical resolution (~2cm) to understand recent distributed accumulation (Trisca et al., 2013; Robertson et al., 2013). The system proved useful for monitoring snow and firn processes but has not been deployed since 2013. These systems set a standard for rover enabled science in the polar regions but remain underused by glaciological

field research teams today due to their expense and the operational expertise required to propose robot enabled polar science and deploy existing polar robots. Other rover deployments in polar regions have been aimed at testing designs of extra-terrestrial rovers, rather than collecting scientific data on the polar regions of this planet. For example, NASA deployed their "Tumbleweed" rover out of Summit Camp, Greenland. These spherical rovers minimize power requirements by relying on near surface wind to transport them across the ice sheet. However, this locomotion mechanism enables serendipitous, rather

than targeted discoveries, as many of the most scientific interesting regions of ice sheets are not likely to be explored by rovers transported primarily by prevailing katabatic winds (Behar et al. 2004). Instrumentation on these rovers was also solely aimed at determining the efficacy of wind transport, i.e. GPS for location and iridium satellite for communication. More recently, autonomous rovers designed by the Chinese National Antarctic Research Program have successfully been used to image crevasses in real time with an intriguing, but unpublished design that includes vertical axis wind turbine power technology.

This rover proved capable of traveling several hundred kilometres on a 58-day deployment but has not yet been used for follow-on missions.

### 1.2 Opportunities Enabled by Modular Rover Design

None of the above-mentioned rovers have been deployed on missions beyond their original project scope (crevasse detection and mapping of internal layers with radar), and the cost to design and deploy polar rovers has remained much higher than the

number of equivalent person hours required to conduct the same volume of work (the rovers often require support of crewed camps). This is principally due to lack of widespread use of rovers beyond the personnel involved in their original design because the rovers are customized, with a seemingly high barrier to use by external personnel. Widespread use and integration of rovers into polar science requires a new approach that motivates our principal rover science goal: to build a modular rover that could be re-purposed to collect observations germane to a variety of polar science programs in the future. This overarching

objective set both our drive design and scientific instrumentation goals. We seek to build a rover capable of >10km traverses on a single battery charge with the capacity to tow 100kg in science payload that could be modified to incorporate a vertical axis wind turbine or solar panel array as the autonomy of the rover is further developed. We also aim to design a rover that accommodates a wide variety of scientific goals, including mapping snow, firn, and ice internal stratigraphy, observing boundary-layer meteorology, and sampling snow. Our rover design is also inexpensive, so that several rovers could be

deployed out of a single camp to collect linked scientific measurements using ground penetrating radar and SfM cameras to conduct surveys of surface and bed topography that are focused on regions where the rovers identify uncharacteristic bed reflectivity or bed roughness. First, we detail the mechanical and software design of the rover system. We then demonstrate the rover's measurement capabilities with a summary of field operations in North East Greenland and conclude by discussing

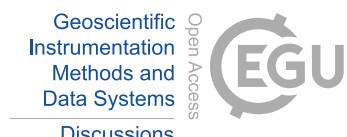

how this system could be used in the future fieldwork to understand feedbacks between surface and bed topography and ice flow via internal layer structure.

## 2 Rover Design

Building a polar rover capable of driving long distances over an ice sheet requires a careful ledger of total traction and
resistance associated with driving in snow. The drive mechanics originally developed for extra-terrestrial rovers have been adapted for use in other polar rover designs to ensure platform mobility and efficiency on snow. We first detail the predicted mobility of our polar rover design by conceptually describing, then calculating, the dominant resistance forces our rover may encounter and the traction we predict for a 100kg system operating in soft snow. We then describe how this design informed build decision making and fabrication. Finally, we summarize the mechanical design with a parameter table comparing
designed and measured rover capability.

### 2.1 Predicted mobility design

We based the mobility design of the CIC rover on Bekker locomotion theory for vehicles manoeuvring over deformable surfaces (Bekker, 1962). The Bekker theory, which was first applied to lunar rover design in the 1960s, aims to represent the wheel-surface contact area to parameterize and understand wheel traction and resistance as a wheel compresses an underlying
surface.  Rover mobility can be characterized by the platform's drawbar pull, or the difference between the rover's surface traction and motion resistance, which defines the ability of the vehicle to move over a given terrain. In the context of ice-sheet exploration, motion resistance acting in opposition to the snow surface traction can be separated into three distinct contributions: work done by the rover to compact the snow surface ($R_c$), internal friction across the drive system ($R_r$; motor inefficiency, friction between the tire and rim hub, bearing friction, etc.), and gravitational resistance encountered when the
rover navigates slopes ($R_g$). Building on the success of existing polar rovers (Lever et al., 2006), we chose to implement a similar design: a 4-wheel drive (4WD) rover over a two track, skid-steer design. The 4WD-in-hub motor system makes the assembly more modular, allowing for more efficient but complicated and expensive two-track systems to be implemented in future iterations of the rover design. To minimize shipping costs and enable deep field deployment with minimal support (all components can be easily moved by a single person), we constrained the total designed rover weight to be less than 100 kg.
We use force-balance analysis of wheel pressure assuming a rover weight of ~ 1000 N equally distributed across four tires (shown in **Figure 1**) to develop equations for resistance from rover sinkage and compaction due to the overburden force on the tire. The parameters used in our analysis are summarized in table 1. We assume, as Bekker did, that snow responds to the applied pressure of a rover wheel as a non-linear spring:



| Parameter | Description (for rover in snow) | Typical value | |
|---|---|---|---|
| $k_1$ | Modulus of snow cohesion | .2 N/cm$^{n+1}$ | |
| $k_2$ | Modulus of snow friction | .1 N/cm$^{n+2}$ | |
| $l$ | Width of wheel | 12.7cm* | |
| $z_0$ | Total depth of sinkage | 1.3cm** | 5 |
| $n$ | Pressure sinkage exponent | .9-1.2 | |
| $D$ | Diameter of wheel | 37cm* | |
| $c_f$ | Internal friction coefficient | .05-.10 | |
| $\phi$ | Snow wheel friction angle | 30-60° | |
| $s$ | Wheel slip ratio. | .02 | 10 |

**Table 1.** Table of parameters used in Bekker locomotion theory and the range of accepted values used to parameterize rover resistance, traction, and mobility. *value specific to CIC rover design. **value follows from choice of parameterization.

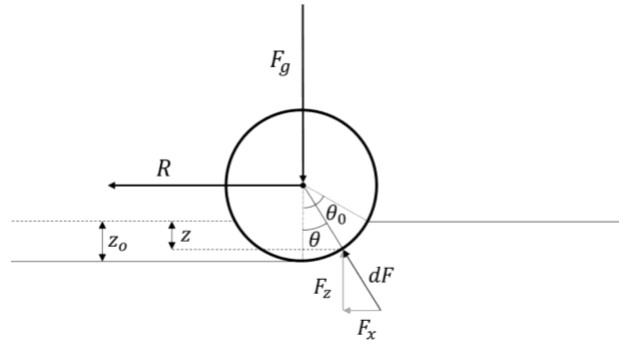

**Figure 1.** Schematic of wheel compressing snow as it drives over a smooth surface. $F_g$ is the overburden force due to gravity acting on the chassis and rover wheel, $R$ is the resistance force felt from compaction and internal friction, $z_0$ is the depth the rover sinks into the ground, and $\theta_0$ and $\theta$ are the angles which become important for integrating the traction around wheel.

20  $$P = \left(k_1 + \frac{k_2}{l}\right) z_0^n$$

where $P$ is the applied pressure from the wheel, $l$ is the width of the wheel, $z_0$ is the final depth that the wheel sinks into the snow, $k_1$ is the modulus of cohesion of snow deformation, $k_2$ is the modulus of friction of snow deformation, and $n$ is a heuristic parameter that captures stress dependence of snow deformation (Bekker, 1962). The force resisting rolling motion of a wheel with diameter, D, due to snow compaction, $R_c$ can be related to the normal force at the wheel snow interface, and then

25  the force of gravity resulting in the equation:



$$R_c = \frac{\left(k_1 + \frac{k_2}{l}\right)}{n+1} z_0^{n+1} = \frac{l \cdot \left(k_1 + \frac{k_2}{l}\right)}{n+1} \left(\frac{3}{3-n} \cdot \frac{F_g}{l\left(k_1 + \frac{k_2}{l}\right)\sqrt{D}}\right)^{\frac{n+1}{n+\frac{1}{2}}} = 15\ N \quad \text{per wheel}$$

with the parameter values assigned in Table 1. Compaction resistance is often assumed to dominate other resistance forces, such as rolling resistance aggregated across all sources of internal friction from the tires and bearings and gravitational resistance encountered when the rover traverses up topographic gradients (Lever et al. 2006). Sastrugi, although common,

rarely exceed slopes of 15°; however, gravitational resistance may become a more important resistance force in more dynamic topographic regions with large scale roughness. For our rover, we assume that rolling resistance, $R_r$ from bearing friction and slip at the wheel-hub interface scales with the weight of our assembly but represents less than 2% of the total resistance and 10% of the gravitational resistance associated with climbing small sastrugi, $R_g$.

$$R_r = F_g c_f = 5N \text{ per wheel}$$

$$R_g = F_g \cdot \sin(\phi_{max}) = 64N \text{ per wheel}$$

where $c_f$ is the total coefficient of friction for wheel and bearing inefficiencies (typically $< .05$), and $\phi_{max}$ is the angle of the slope the rover climbs as it moves across the ice sheet. In our design, we neglect bulldozing resistance, which occurs as snow is displaced in front of the rover's advancing wheels and depends on snow cohesion and wheel geometry. Narrow wheels reduce bulldozing resistance by significantly increasing the ratio of snow that is displaced to the sides of the rover wheel versus

the snow that is pushed forward in front of the wheel; however, with narrower wheels the rover sinks deeper into the snow. Because the sinkage depth of the wheels relative to the wheel diameter is greater than 6%, we can assume that bulldozing can be neglected in the force-balance calculation (Ishigami et al., 2007).

For a 4-wheel model, we anticipated that the dominate resistance forces come from wheel sinkage on the front wheels and

gravitational resistance as the rover traverses in and out of the shear-margin troughs of the NEGIS, where steeper surface slopes occur. A conservative estimate of the total resistance force is:

$$R_{tot} = \left(R_c + R_r + R_g\right) = 326\ N$$

The dimensionless resistance parameter our rover must overcome to move across the most extreme conditions of the Greenland ice sheet interior is

$$\frac{R}{F_g} = 0.33$$

This value is noticeably larger than the estimates produced by Lever et al. (2013) and Ray et al. (2007) due to the higher estimate for gravitational resistance and the heavier anticipated weight. This value represents an upper bound on the resistance forces, as we will see when we compare it to values measured during the field deployment. The semi-empirical Bekker locomotion model also uses the relationship between the tire and snow surfaces and the shearing strength of the tire motion to

predict vehicle mobility. The shearing strength at the wheel snow interface can be described by the horizontal rover traction:

$$R = (A \cdot C + F_s \tan\phi) \cdot \left(1 - \frac{K}{l}\left(1 - e^{-\frac{sl}{K}}\right)\right)$$

where $A$ is the contact area of the tire with the snow, $C$ is the snow-wheel cohesion, $\phi$ is the snow-wheel friction angle, $s$ is the wheel slip ratio, $K$ is the coefficient of slip, $l$ is the length of the contact path, and D is the diameter of the wheel, $\frac{D}{2} \backslash\cos^{-1}\left(1 - \frac{2z}{D}\right)$. Values for these parameters come from vehicle tests at Summit station (Army Corps of Engineers, 1972).

The dimensionless form of the drawbar pull, DP, which captures the effective traction, and the load we could pull with the rover takes the form

$$\frac{DP}{F_g} = \frac{T - R}{F_g} = 0.29$$

where $T$ is the traction at the snow-wheel interface and $R$ is the total resistance, and $\frac{DP}{F_g}$, the dimensionless efficiency metric of the rover we calculate using parameters for the predicted CIC rover design. The predicted drawbar pull suggests that the CIC rover can pull a sled with 80 kg of freight using just 120W of power, more than adequate cargo capacity to tow a ground penetrating radar at least 10 km under the efficiency and energy density constraints of inexpensive lead acid batteries.

**2.1 Chassis and Wheel Design**

The CIC rover design is also compact, because the entire system was built to ship inside two $80 \times 60 \times 61$ cm shipping containers (Zarges, 2017). We designed the chassis to come apart in four pieces that can be assembled into the rover body and an accompanying lid that protects the electronics, motors, and controllers from the external environment. Stainless-steel axles were cut with collars that interface directly with four brushless DC motors using dual screw keys. These collars protect the fixed rover motors from bearing radial torques that could damage the motor shafts. Coupling the motors to the stainless-steel

axles are four planetary gears with gear ratios of $1:77$, which were chosen to produce sufficient torque at the wheel to overcome all predicted resistance forces. Because of the cost and limited redesign capabilities of standard wheel-rim-axle assemblies, we printed polyethylene rover rims with the Ultimaker 3 extended 3D printer. These rims hugged four youth ATV snow tires that remained at carcass pressure with a wheel diameter of 0.38m. The four brushless DC motors propel the rover, operating near their max speed of 3000rpm with a nominal torque of 0.4Nm before the planetary gearbox. After the gearbox,

the motors supply a nominal torque of 26Nm rotating at 39rpm. Assuming that the tires deform to effective diameter of $\sim$ 36cm under the weight of the rover, the top speed of the rover is:

$$V_{max} = \frac{39rpm}{60sec} \cdot 2\pi \cdot .19 = 0.77\frac{m}{s}.$$

The motor controllers are pulse width modulation servo amplifiers fixed using 5V square wave signals across analog pins of an Arduino MEGA microcontroller (Arduino 2016). Under-voltage (below 18V) and overheating (above 95°C) emergency

switches protect the controllers and override the Arduino in case of electrical load depletion or internal malfunction. The rover



signals the motor controllers from the Arduino MEGA to navigate. The rover turns by reducing the rotation speed of the wheels on the same side of the chassis as the direction of the turn. The movement is equivalent to the skid steering of a tracked vehicle (i.e snow mobiles) except that the wheels themselves are coupled via the speed controller signalling, and not mechanically linked with treads.

Besides our choice of motors, the drive system we designed was also built to operate in the low temperatures found in polar regions. Using grey Styrofoam, we insulated two 12V, 70amp hour sealed lead acid gel-cell batteries lined in series to power the assembly, keeping the lead acid batteries above freezing at all times to enhance battery efficiency. All other rover parts were purchased to perform at their designed specifications in temperatures as low as $-10°C$; however, tests during the summer

10 in Greenland proved that the rover was fully operational at temperatures as low as $-23°C$.

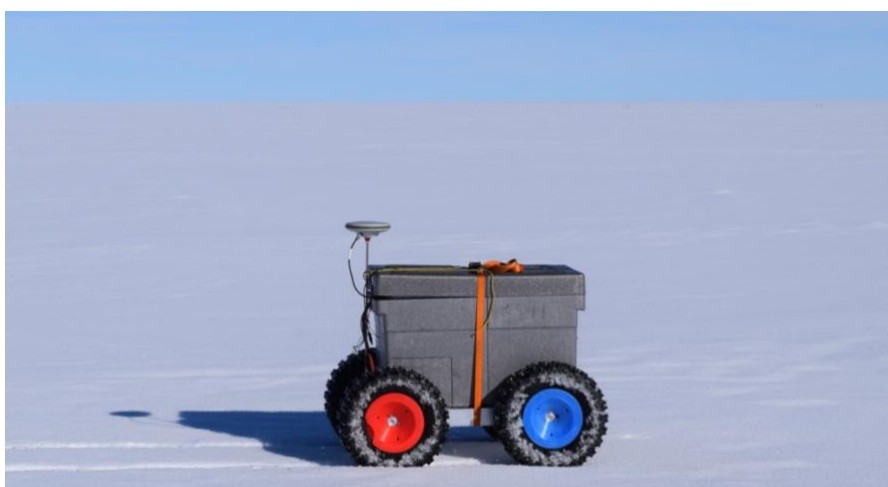

**Figure 2.** Photograph of the CIC rover taken on July 4[th], 2017 during its first mission out of the EGRIP camp.

| Rover Specifications | | |
|---|---|---|
| Parameter | Design Specification | CIC Rover |
| Rover Mass | <100kg | 85kg |
| Pulling Force (N) | 300 | 250 |
| Dimension (m$^3$) | <1 | $0.7 \times 0.7 \times 0.6$ m (0.3 m$^3$) |
| Wheel radius (cm) | 20 | 18 (fully inflated) |
| Speed (m/s) | - | .77 |
| Run time (hours) | 8 | 12 |
| Path Tolerance (m) | +/- 5 | +/- 7 |



| | | |
|---|---|---|
| Minimum Temperature (ºC) | -20 | -23 (lowest temperature encountered) |
| Dimensionless Resistance R/W | 0.33 | 0.14 |
| Cost | $3000 | $2900 |

**Table 2.** Summary of CIC rover design parameters and measured performance in the field.

### 2.2 Software, navigation and Instrumentation

During a mission, the rover may be directed to make and log several different measurements. The navigation and measurement sampling algorithms must have the ability to set appropriate different priorities during these missions. As the rover powers, the microcontroller first runs an initializing set of commands that are contained within a C-programmed initialization script. These commands establish connections between the microcontroller and the GPS module and the micro-SD card, test that all motors and motor controllers are connected, and calibrate the power sent to all wheels for the rover to drive in the straightest path. The initialization script is followed by the primary loop that executes navigation commands and communications with external instruments and drive system hardware. The structure of the primary loop follows a hierarchy of hardware checks and instrument sequences centred around a query to the onboard GPS to establish rover position every second. A serial port military connector in the chassis rear wall allows for manual rover control, while a dual-frequency GNSS receiver (Leica GNSS 1200) allows the rover to navigate to waypoints loaded to the rover via the micro-SD card prior to initialization. In the main loop of the navigation and instrument triggering algorithm, an interrupt key can be pressed to switch between autonomous drive mode and user navigation mode, which is the default to start. During a cold start, the receiver must parse time, position, and almanac clock corrections from each satellite, so for an initial run, the rover waits five minutes to constrain its initial position before manoeuvring to the first waypoint. Initially, while the rover is manually controlled, turning, rover acceleration, and instrument triggering are all initiated across the USB-port. During an autonomous mission, the rover uses a haversine algorithm to calculate the distance and bearing from the rover's current position to the next waypoint. The distance to the next waypoint can be determined from the rover's current position $(lat_1, lon_1)$ and the proximal waypoint position $(lat_2, lon_2)$, and a calculation of the bearing in degrees from North follows from the distance to the next waypoint:

$$distance = 2 \arcsin \sqrt{\cos(lat_1) \cdot \cos(lat_2) \cdot \left( \sin^2 \frac{(lon_1 - lon_2)}{2} \right) + \sin^2 \frac{(lat_1 - lat_2)}{2}}$$

$$bearing = \arccos \frac{(\sin(lat_2) - \sin(lat_2) \cdot \cos(d))}{\sin(d) \cdot \cos(lat_1)}$$

The navigation bearing is the corrected course the rover travels based on the great circle path from the rover's current position to the next waypoint using the rover's current position and a linear average of the rover's bearing since the last correction. From the corrected bearing and distance to the next waypoint, the rover turns towards the waypoint if the difference between the current bearing and the corrected bearing exceeds 5°. In both manual and autonomous drive modes, the rover turns by





signalling motors synchronously on the turn-direction side of the chassis to slow down by a pre-set percentage of the drive speed depending on the bearing difference. In autonomous drive mode, the rover derives a bearing from the difference between the two previous positions every 3 seconds. This GPS positioning information is also used to fit a line for rover velocity from the 10 previous GPS readings. If the velocity from the linear fit to displacement is less than the expected value based on

signalling from the motor controllers, the rover will slow down. The microcontroller will eventually re-signal the motor controller to speed up when the velocity of the rover is consistent with the measured velocity from GPS ensuring that if the wheels begin to slip at the snow interface, the rover can slow down and regain traction. When the rover reaches the final waypoint, it stops powering the drive assembly, but continues to take GPS measures that could be used in a radio telemetry script to expedite rover retrieval. The measurement sampling frequency and GPS turning correction frequency are initialized

in the setup script to user input values. For the CIC rover, they were set at 3 second and 20 seconds, respectively, and were based on requirements for the SfM photogrammetry analysis carried out at the end of July.

The software and instrument design we have chosen is also modular and could be changed as we further develop the rover's autonomous capabilities. For example, the navigation protocol presently consists of uploading predetermined waypoint paths

directly to the system before the rover embarks on a mission. This simple procedure could be expanded to incorporate live instrument data (i.e. surface photos, radar imagery) as real-time input into an artificially intelligent navigation system that would learn and adapt to a specific research setting and science objective. As proof of concept for the rover's autonomous capabilities, we next present a workflow for automated ground-based SfM photogrammetry, which in future iterations of the rover design could be used to build real-time elevation maps that could be incorporated into rover navigation.

**2.2 Instrumentation: Close-range photogrammetry**

We chose not to design instruments specifically for rover deployment. We hope to provide a platform that can support different instrumentation for a variety of scientific goals and seek to develop that framework instead of creating instrumentation solely for science specific to EGRIP. For our test deployment, we focused on developing the ability for surface elevation mapping via structure from motion photogrammetry, which could serve many scientific and operational goals, by installing a dual-

frequency GPS receiver and camera on the rover. The SfM camera selected for initial field testing of the rover instrumentation camera was mounted and wired to the microcontroller in the field and is primarily meant to demonstrate the signalling capabilities of the rover and the platform's utility for linked surface and atmosphere studies. Scientifically, there is a long history of measuring relative snow and ice elevation using photogrammetry. Baltsavias et al. (2001) provide a historical review on glacier monitoring and used automated digital elevation model extraction tools based on photogrammetric area correlation

to generate elevation models of Unteraargletscher, Switzerland. The photogrammetry digital elevation model matched one derived from airborne laser scanning data, except in areas where the spatial contrasts and variations in colour and light intensity were low. At closer range, Kaufmann et al. (2013) applied photogrammetry to time series of historic terrestrial photographs to monitor glacier ablation at Goessnitzkees, Austria, with decimetre uncertainties. In the ice-sheet interior, centimetre-scale



elevation maps of surface roughness can be used to better understand sastrugi evolution and, coupled with ground penetrating radar (GPR) mapped snow stratigraphy and water isotope measurements, to better understand the influences of surface topography on snow densification, ice flow, and surface isotope deposition and fractionation. In low accumulation environments, such as EGRIP, snow bedform evolution may control spatial variations in accumulation and isotopic

composition of snow and firn near the surface. Over long timescales (and related depth-age sales in the ice) these signals are smoothed but could have implications for interpreting shallow ice-cores. For these reasons, the rover was taken to the EGRIP camp for field evaluation, which may lead to its integration into future surface science projects.

## 2.2 Field Performance

Rover performance was evaluated in North Eastern Greenland (Figure 3) where we tested the rover's dimensionless resistance,

the signalling capabilities for instrument measurements, the total distance the rover could travel, and the resolution of rover-photogrammetric methods. The rover was deployed out of the EGRIP camp during the month of July 2017. During this time, three precipitation events allowed us to test the rover's performance in both drifted and compacted snow conditions. The rover successfully completed four 20km missions from camp, navigating within 7m of prescribed waypoint path. Inefficiencies in the rover directional control due to motor controller current drift forced subtle drive corrections every 20m to keep the rover

on track, which made the rover's actual travel path between waypoints, at times, 10% longer than the direct path. Coupling angular frequency referencing measurements at the wheel with the signalled speed controllers would easily correct this problem, but the necessary spare parts were not available during the summer 2017 deployment. We performed several traction tests to evaluate the predicted rover resistance against measured values. We carried out these tests by loading a sled with different weights and measuring the towing force with a digital spring scale. We found that, on average, the rover encounters

a dimensionless resistance, $\frac{R}{F_g} = 0.14$, that reached a maximum value $\frac{R}{F_g} = 0.4$, when driving over a field of rippling sastrugi.

The rover was able to pull three cargo boxes and a person, a weight of nearly 170kg, in a low friction plastic sled in untouched snow. This weight is more than twice that of common commercial ground-penetrating radar systems, which could be deployed on the rover to map accumulation layers.



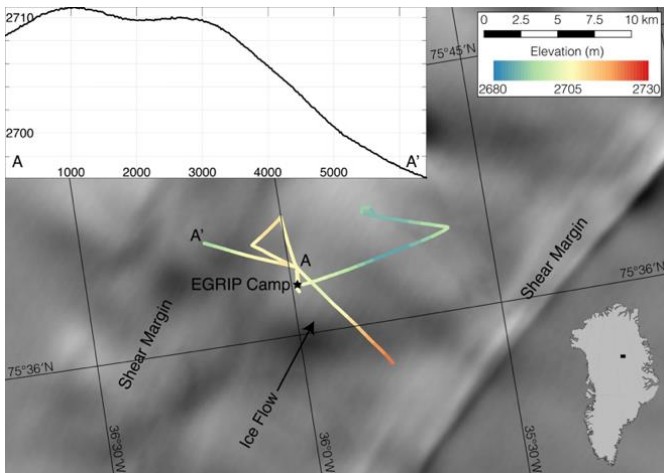

**Figure 3.** MODIS image of North East Greenland Ice Stream and the East Greenland Ice Core Project Camp with rover mission paths coloured by surface elevation of the ice as determined from the CIC rover GPS.

By the end of the July field period, the internal width of the tires of the rover had shrunk, so that, in drifted snow, the force needed to overcome the frictional force of sinkage became greater than the traction between the tire carcass and the wheel rim. Occasionally, this would cause the wheel rim to spin freely inside the tire carcass, wearing away part of the tire and exacerbating the tire-rim misfit. Because of our 3D printable rim design, new wheel rims could be printed in the field that were

1 cm wider, increasing the traction between the hub and the tire wall, and allowing the rover to continue working after one day of inaction during wheel rim printing.

To evaluate the capability of the rover to autonomously take photographs that could be subsequently used to create digital elevation models via structure-from-motion photogrammetry, we installed a Nikon D3000 DSLR camera to the top of the

chassis. We signalled the camera from the micro controller using one of the Arduino MEGA's four serial channels. Because the instrument sampling frequency for the SfM camera and drive correction interrupt frequencies are so much smaller than the quartz oscillator on the microcontroller, the instrument signalling does not conflict with navigation and is unimpaired by the processing speed of the microcontroller. Instruments that require high precision triggering, such as radar receivers, would need to externally stack and log data in order to prevent mistriggering pulses from the Arduino, which cannot write fast enough to

trigger, record, and stack received radar pulses.

↓   ↓





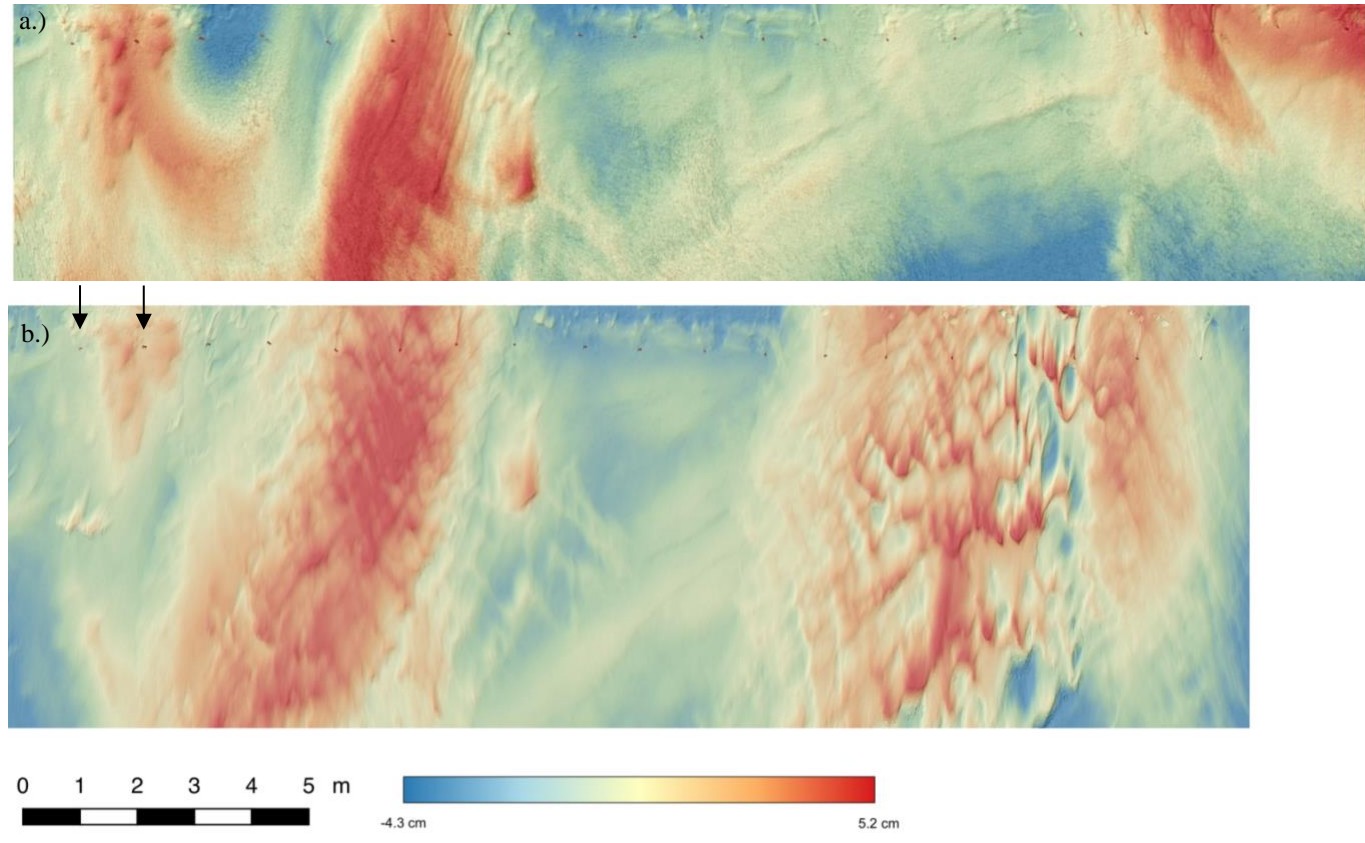

**Figure 4.** Relative snow surface height in the region monitored for accumulation. Panel (a) shows surface height on July 19th before a storm system swept through the EGRIP camp on July 20$^{th}$, and panel (b) shows surface height after the storm on July 22$^{nd}$. Black arrows in upper left corner of each panel indicate the first two poles in the row of accumulation monitoring sites.





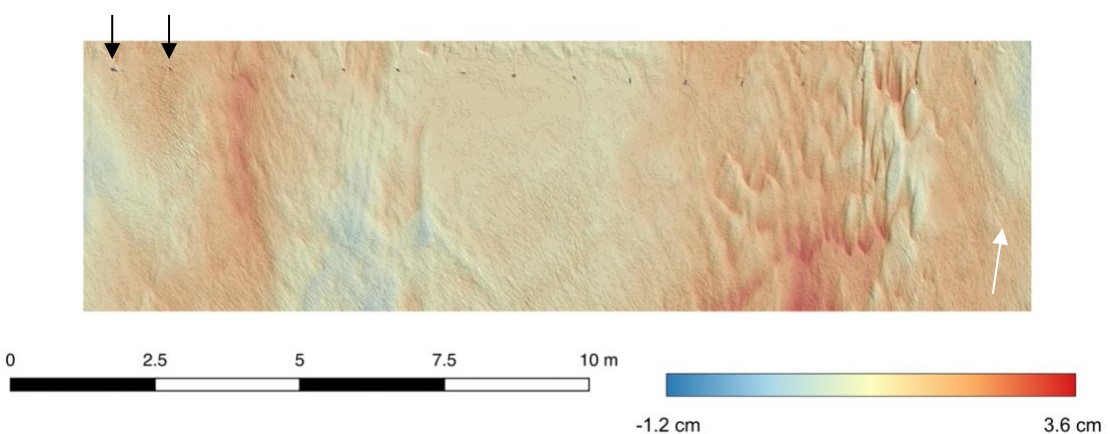

**Figure 5.** Snow height difference between elevation models from July 19th and July 22nd measured from the top of the poles inferred to be accumulation during the storm. Black arrows indicate the first two poles in the row of accumulation monitoring sites. White arrow indicates prevailing wind direction during the storm.

The Nikon D3000 camera mounted to the rover was triggered to take one picture every second as the rover passed by a region outside of the EGRIP camp that was manually monitored for accumulation by measuring elevation height change relative to 200 wooden poles. We used these photos to construct digital elevation models of the snow surface vis SfM photogrammetry. SfM photogrammetric processing triangulates the position of co-imaged snow features from multiple photographs to create a Cartesian point cloud of the measured surface. The point cloud can then be gridded into a 3D relative digital elevation model (DEM) by interpolating different samples of points and comparing their misfit to create a map of relative elevation and elevation uncertainty. We used Agisoft's Photoscan structure-from-motion (SfM) algorithm to process 50 photos taken by the rover in the first 25m of this site on July 19th and July 22nd (Koenderink and Doorn, 1991; Westoby et al., 2012). From the Lagrangian reference frame of our accumulation poles, we derive maps of relative surface height and surface height change before and after a storm swept through the area on July 20th - July 21st in **Figure 5**. We report our precision as $+/- 95\%$ of the root-mean-square error of the elevation difference between the two digital elevation models (DEM) after they have been optimally co-registered to the accumulation poles. Overall, these maps measure relative elevation with an accuracy better than 1cm and precision better than 4mm. To ensure that the difference along the stake row is faithful to the existing measurements for accumulation, each stake is compared to stake heights measured manually, which match within the tolerance of error except where the poles created significant depressions in the surface and the relative height of the surface to the top of the pole became more ambiguous. The distribution of poles is not completely uniform, so we can use the misalignment of the row of poles to our advantage to find the single affine translation that best aligns the elevation models. We then difference our DEMs to determine accumulation due to drifting and snow fall between the two image acquisition periods.





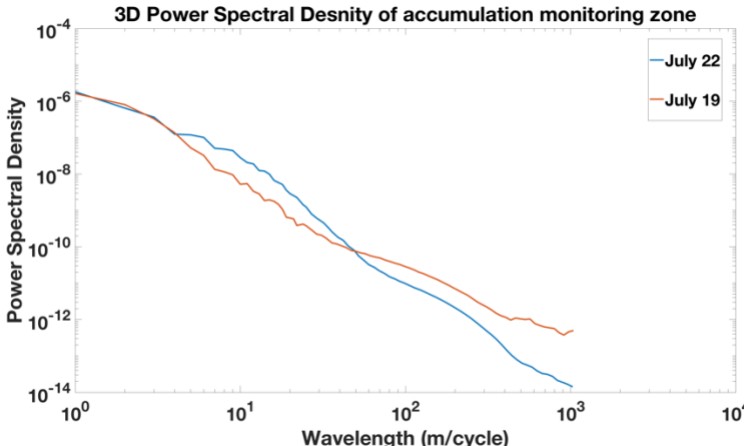

**Figure 6.** Power spectral density periodogram of the region of snow accumulation taken from the rover before and after the storm. The power changes from low frequency to high frequency after the high wind speeds sculpt steeper, shorter wavelength

sastrugi.

After the storm, we find higher amplitude finer wavelength features (0.5-2m). These small-scale features inevitably diffuse away as wind speeds fall below the threshold velocity required to maintain the steeper slopes of small wavelength features (Filhol and Sturn, 2015) so that the surface will eventually resemble the one we recorded before the storm. Wind speeds rose

from 0-2 m/s before the storm to over 10 m/s during the seven-hour storm, causing a transition in dominant snow grain transport process from creep-dominated snow bedform evolution to saltation of grains and the formation of finer sastrugi. In future iterations of this low-cost rover, real-time processing of atmospheric data could feedback into the drive system control software to allow adaptive driving over an evolving surface with a roughness that is highly dependent on the recent precipitation and local wind speed history (Filhol and Sturn, 2015).

In our photogrammetric analysis, we assume that the poles set out to monitor accumulation do not move relative to one another and represent faithful markers of unchanged position in the Lagragian reference frame of the moving ice. This assumption is already being made in the manual measurement protocol, which uses changes in height from the top of the stake to the surface to infer accumulation change. However, in our elevation maps, we see that the poles themselves change the pattern of local

surface height so that relative elevation measurements at the poles cannot resolve millimetre-scale changes in surface height. Outside of the region monitored for accumulation, where there are no snow free markers to difference elevation models, we can use the rover precision GPS to establish referenced camera positions and camera angles. In this framework, map precision depends on the co-registration of features across photos and also on the uncertainty of the camera position calculated from the GPS. The close proximity of the camera to the imaged surface and the stability of the camera-GPS-rover platform make the

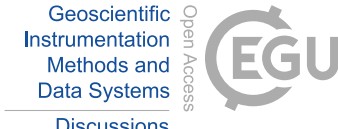

surface height model an excellent complement to satellite and winged-drone derived elevation models for high-resolution targeted surveys.

## 3 Discussion

A class of inexpensive, modular rovers based on the design we have developed would be simpler and less expensive than existing polar rover designs, which we hope will lead to more widespread adoption of this platform in polar field programs. The simple design permits easy customization for wide use outside geophysical glaciology. Because the platform is ground-based, it is extremely stable, and can carry over 20 kg of additional instrument weight in its chassis. The rover is also emission free, allowing it to cleanly sample the near surface boundary layer important for quantifying water vapor exchanges between the snow surface and the atmosphere, and to sample the snow surface directly for snow microstructure and snow specific surface area measurements. Because the time scales of surface processes, such as grain saltation and molecular diffusion, are so short, an instrument that cleanly samples the near surface atmosphere and snow at one location simultaneously is vital for characterizing the effects of these processes on snow water vapor isotopes and snow density.

The low cost of the rover also enables design of projects that use multiple rovers simultaneously. With the addition of UHF inter-rover radio communication, these rovers could be coordinated and controlled to create high-resolution maps of strain patterns on the ice-sheet surface (via GPS surveying) and internal layers (via ground-penetrating radar) along repeatable tracks. A linked network of rovers equipped with ground-penetrating radar would allow real-time radar survey redesign to densify sampling in areas of high uncertainty, such as where the bed reflector becomes discontinuous or dims or where cross-track or off-nadir reflections between rovers disagree. Because of their low cost, the rovers are also expendable, and could be sent to image internal ice layers and measure surface strain rates in some of the most dynamic regions of the ice sheet that currently are undersampled because of crevassing. In the future, we plan to use the rover to make repeat radar measurements to improve understanding of how ice flow affects the internal layers of the ice sheet in regions of fast flow, such as the North East Greenland Ice Stream shear margins. We are also building a mountain glacier rover model for exploration of Cascadia glaciers and remote semi-autonomous earth science research more broadly.

## 4 Conclusions

The development of the CIC rover was motivated by our goal to design an autonomous rover platform for hazardous and power-restricted remote sensing of the Earth's ice sheets. This set our design considerations: to make a research platform that will conserve field resources and extend the range of data collection to dangerous, isolated ice-sheet regions that go undersampled due to the threat of crevasses or severe weather. We have demonstrated the versatility, durability, and value of the CIC platform through test deployments in northeast Greenland out of the EGRIP camp. When our printed parts (chassis and rims) were damaged, a repaired rover using new printed parts was redeployed after just one day. Components that cannot be printed (batteries, motors, microcontrollers) are modular, allowing for easy replacement, if damaged, and specialization



depending on rover use. Future rover design efforts will focus on adding a solar panel array and charging station that leverages the Greenland summer's solar power availability to extend the rover's range from tens to hundreds of kilometres, and the integration of instruments into a now proven rover platform. The rover's success demonstrates that semi-autonomous polar instruments can be built at very low costs (<$3000 USD), opening up new measurement campaign possibilities that involve

several platforms that conduct linked surface science simultaneously.

## Acknowledgements

This work was supported by the research grants (2361 and 16572) from VILLUM FONDEN. Logistical support was provided by the EastGRIP project, which is directed and organized by the Centre for Ice and Climate (CIC), Niels Bohr Institute, University of Copenhagen. We thank Karl-Emil and the NBI workshop for their help in the rover-design process. For queries

concerning the accumulation dataset and the latest navigational code with instrument procedurals contact Andrew Hoffman: hoffmaao@uw.edu

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
