# Peer review of "A low-cost autonomous rover for polar science"

_Geoscientific Instrumentation, Methods and Data Systems, 2018_

## Referee Comment (RC1) · Anonymous Referee #1 · 24 Jan 2019

In this paper the authors provide the technical specifications and a feasibility study of a low-cost autonomous rover designed to operate in hostile environments. Also, the authors describe the performance of the proposed solution on real missions. The first part of the manuscript is quite detailed, but the discussion must be improved. I think that the manuscript can be considered for publication after major revisions as specified below:

- The paragraph Field performance (experiments) should be extracted from the Rover design section and integrated in an "Experiments and results" dedicated section. This new section should be expanded as specified in the next bullet point.

- A full comparison with at least two different solutions currently available should be given, as well as an analysis of possible upgrades/downgrades of the proposed solu-

tion, for example mounting different imaging sensors (less or more expensive).

- I believe the manuscript is quite unbalanced. I missed a more comprehensive discussion section. It is clear that the developed system is simpler and less expensive with respect to other commercially available products, but I think that the possible drawbacks of the solution should be highlighted as well. Also, examples of additional instruments that can be carried on the rover should be given. The implications of using the proposed solution instead of other commercially available ones need to be clarified and explicitly written.

- In the conclusions the authors write about future efforts for the integration of solar panels. This fact should be clearly addressed also in the discussion, for example highlighting the possible costs for specific add ons. In fact, the total cost of the rover can not be the same of the "base" model integrating other kind of instrumentation.

- A minor remark: please pay attention to section numbering (e.g. 2.2 field performance has the same number of the previous subsection) and

---

## Referee Comment (RC2) · Schmidt (Referee) · 2 Feb 2019

General comments

The paper "A low-cost autonomous rover for polar science" by Andrew O. Hoffman, Hans Christian Steen-Larsen, Knut Christianson, and Christine Hvidberg describes the design, implementation and test of a light-weight and inexpensive autonomous rover for polar research. According to the authors it can be easily adapted for a wide range of scientific objectives and makes areas accessible which are difficult or expensive to explore using human field operatives. Due to the simple design it might be deployed without highly specialized know-how only available to the original development team like most previous rover missions. The article is well written and contains sufficient

details to allow the reader an evaluation as to the concept's usefulness for own application areas. Though the design focuses on a rover deployment on polar ice sheets, different applications in harsh environments seem to be feasible where direct human access might be difficult or dangerous.

Specific comments

Page 4, table 1: For what temperatures and/or snow densities are the given modulus values valid? Snow coherence and surface friction are heavily dependent on these environment conditions. As these conditions determine via the equation on line 20 under which conditions the design can be safely used, a rough estimate of the pressure under considered extreme conditions would be helpful.

Page 8, line 7-8: passive insulation to keep the battery above freezing? Is active, battery-powered heating needed to ensure this temperature at very low outside temperatures and intermittent low-power usage? See remark in the introduction of the article hinting at operation also under polar winter conditions.

Controller and electronics temperatures? What are the lowest temperatures the components of the control electronic can operate in? Does the electronics compartment need heaters to start and/or operate under very low external temperatures? Low temperature components are often built to military specifications and are expensive or restricted for political reasons to certain applications and owner countries.

---

## Author Comment (AC1) · 4 Apr 2019

Title: A low-cost rover for polar science

Journal: Geoscientific Instrumentation, Methods and Data Systems

Authors: Andrew O. Hoffman, Hans Christian Steen-Larsen, Knut Christianson, Christine Hvidberg

Article Number: gi-2018-52

Date: 26 March 2019

In our manuscript, A low-cost rover for polar science, we present the developmental considerations, design, and deployment of an autonomous modular terrestrial rover for

ice-sheet exploration that is inexpensive, easy to construct, and allows for instrumentation customization. The primary goal of the study was to demonstrate that science groups can build their own rover solutions to address different multi-scale data acquisition needs in polar science. We thank both reviewers for a critical reading of our paper. Reviewer comments highlighted three manuscript weaknesses with varying degrees of specificity: (1) expansion of the discussions section to compare our work to that of others, (2) additional costs to outfit a rover with instrumentation that are not detailed here, and (3) some users may have difficulty constructing rovers due to lack of access to components in some countries.

Here, we address these comments and provide a point-by-point response to more specific referee criticisms. Below, the reviewer's comments are marked in normal text and our responses are in bold. We mark new line numbers for the revised manuscript with the prefix R and original lines numbers with no prefix.

Referee #1

In this paper the authors provide the technical specifications and a feasibility study of a low-cost autonomous rover designed to operate in hostile environments. Also, the authors describe the performance of the proposed solution on real missions. The first part of the manuscript is quite detailed, but the discussion must be improved. I think that the manuscript can be considered for publication after major revisions as specified below:

- The paragraph Field performance (experiments) should be extracted from the Rover design section and integrated in an "Experiments and results" dedicated section. This new section should be expanded as specified in the next bullet point.

We have followed the reviewer's organizational suggestion (p11 R14 - p14 R23). The new section is now numbered (p11 R14).

- A full comparison with at least two different solutions currently available should be
given, as well as an analysis of possible upgrades/downgrades of the proposed solution, for example mounting different imaging sensors (less or more expensive). I believe the manuscript is quite unbalanced. I missed a more comprehensive discussion section. It is clear that the developed system is simpler and less expensive with respect to other commercially available products, but I think that the possible drawbacks of the solution should be highlighted as well. Also, examples of additional instruments that can be carried on the rover should be given. The implications of using the proposed solution instead of other commercially available ones need to be clarified and explicitly written.

We tried to more clearly emphasize the differences between our new rover and previous designs (p15 R14 - p16 R27). We have rewritten the discussion section highlighting advantages and disadvantages of low-cost rovers in polar science (p15 R23 - p16 R26), but we unfortunately cannot compare our rover directly with other rovers because 1) the few commercially available rovers that exist are poorly suited for deployments in polar environments making for an unfit/unfair comparison and 2) the rovers developed by other groups have been proposed with highly specific research objectives in mind highlighted, as highlighted in section 1.1. 3. We now clearly state that this makes easily adapting other rovers to polar fieldwork difficult (p3 R20). Existing rovers are also not necessarily available to all groups and we directly state this (p15 R20). We also now more directly address the weaknesses of the polar rover and frame future development around improving upon the low-cost modular rovers' weaknesses (p16 R29 - p17 R10).

In the conclusions the authors write about future efforts for the integration of solar panels. This fact should be clearly addressed also in the discussion, for example highlighting the possible costs for specific add ons. In fact, the total cost of the rover cannot be the same of the "base" model integrating other kind of instrumentation.

We thank the reviewer for noticing this oversight and now discuss possible revisions to the power system in the discussion (p16 R21). We admit that the total cost of a rover fitted with new instrumentation and redesigned power system could be substantially more than the low-cost platform described. However, we wish to provide a flexible platform that could be adapted to suite various budgets, and now explain this point more clearly (p3 R18). To improve overall organization and emphasize additional developmental considerations, we have added a second discussion section (p16 R30) describing future iterations of rover development wherein we outline improvements that acknowledge the increase in total cost of the rover associated with modifying the power bus and the communication software (p16 R30 – p17 R4).

A minor remark: please pay attention to section numbering (e.g. 2.2 field performance has the same number of the previous subsection) and

This has been corrected (p4 R9, p6 R18, p9 R3, p10 R26). We have checked all other section numbers and do not believe there are additional mistakes, but apologize if we missed some.

  Referee #2:

General comments The paper "A low-cost autonomous rover for polar science" by Andrew O. Hoffman, Hans Christian Steen-Larsen, Knut Christianson, and Christine Hvidberg describes the design, implementation and test of a light-weight and inexpensive autonomous rover for polar research. According to the authors it can be easily adapted for a wide range of scientific objectives and makes areas accessible which are difficult or expensive to explore using human field operatives. Due to the simple design it might be deployed without highly specialized know-how only available to the original development team like most previous rover missions. The article is well written and contains sufficient details to allow the reader an evaluation as to the concept's usefulness for own application areas. Though the design focuses on a rover deployment on polar ice sheets, different applications in harsh environments seem to be feasible where direct human access might be difficult or dangerous.

Specific comments

Page 4, table 1: For what temperatures and/or snow densities are the given modulus values valid?

The parameter selection followed from a similar treatment from Lever et al. (2006), which is cited, and we added a lengthy description of the parameter values in the table footnote (p5 R12 - R18). As we are mainly following the work of others here, we believe that a citation should be sufficient reference. We do, however, note that we found these values to work well in our field test of this rover in northeast Greenland.

Snow coherence and surface friction are heavily dependent on these environment conditions. As these conditions determine via the equation on line 20 under which conditions the design can be safely used, a rough estimate of the pressure under considered extreme conditions would be helpful.

The reviewer is correct that snow coherence and surface friction change with environmental conditions. However, the overburden pressure of the rover assembly does not change. There is some change in stress with depth due to variable snow conditions, but we focus our very primitive analysis on the deformation of the snow as this is what can be linked to a work calculation required to deform paths created by the rover as it moves over the snow. The response of the snow only changes as a function of snow parameter selection as we keep the rover weight constant. We try to clarify these points in lines p5 R15 - R16.

Page 8, line 7-8: passive insulation to keep the battery above freezing? Is active, battery-powered heating needed to ensure this temperature at very low outside temperatures and intermittent low-power usage? See remark in the introduction of the article hinting at operation also under polar winter conditions. Controller and electronics temperatures? What are the lowest temperatures the components of the control electronic can operate in? Does the electronics compartment need heaters to start and/or operate under very low external temperatures?

The batteries noted by referee #2 were able to operate under very low external temperatures because of the insulative styrofoam sealed carriage, which kept the batteries above freezing at all times. However, there are numerous lead sealed acid batteries capable of operating with only marginal performance degradation for temperatures as low as -40°C. The electronics stored in the chassis with the motors and motor controllers were not heated (expect for minor fictional heating form the motors themselves). Wider temperature range tests in a cold room (that would still allow for the onboard GNSS receiver to telemeter with the GNSS satellite constellations) or during the seasonal transition from summer to winter were outside the scope of this original study, but would make the arguments for polar tested platform dependability stronger. We have noted now that additional tests of performance under extreme temperatures would be valuable (p8 R16).

Low temperature components are often built to military specifications and are expensive or restricted for political reasons to certain applications and owner countries.

The connection cabling was the only part most commonly used by national militaries, but these connectors are widely available on large-scale distributor sites such as digikey and Fastenal, and are built to military specifications rather than restricted by country origin. On the topic of costs and unequal access, all parts were purchased and shipped to Denmark, which charges a much higher fee for imported goods than the United States, for example, and many other countries. The costs we list are what we paid and have not been corrected to account for import duties, taxes, or other fees (noted in p15 R21).
* * *